# Composition of nitrogen in urban residential stormwater runoff: Concentrations, loads, and source characterization of nitrate and organic nitrogen

**Jariani Jani[1], Yun-Ya Yang[2], Mary G. Lusk[3], Gurpal S. Toor[2]***

**1** Chemistry Department, Faculty of Science, University of Malaya, Kuala Lumpur, Malaysia, **2** Department of Environmental Science and Technology, University of Maryland, College Park, MD, United States of America, **3** Gulf Coast Research and Education Center, University of Florida, Wimauma, FL, United States of America

* gstoor@umd.edu

**Data Availability Statement:** All relevant data are within the manuscript and its Supporting Information files.

**Funding:** GST received funding from Florida Department of Environmental Protection. There is

## Abstract

Stormwater runoff is a leading cause of nitrogen (N) transport to water bodies and hence one means of water quality deterioration. Stormwater runoff was monitored in an urban residential catchment (drainage area: 3.89 hectares) in Florida, United States to investigate the concentrations, forms, and sources of N. Runoff samples were collected over 22 storm events (May to September 2016) at the end of a stormwater pipe that delivers runoff from the catchment to the stormwater pond. Various N forms such as ammonium ($NH_4$–N), nitrate ($NO_x$–N), dissolved organic nitrogen (DON), and particulate organic nitrogen (PON) were determined and isotopic characterization tools were used to infer sources of $NO_3$–N and PON in collected runoff samples. The DON was the dominant N form in runoff (47%) followed by PON (22%), $NO_x$–N (17%), and $NH_4$–N (14%). Three N forms ($NO_x$–N, $NH_4$–N, and PON) were positively correlated with total rainfall and antecedent dry period, suggesting longer dry periods and higher rainfall amounts are significant drivers for transport of these N forms. Whereas DON was positively correlated to only rainfall intensity indicating that higher intensity rain may flush out DON from soils and cause leaching of DON from particulates present in the residential catchment. We discovered, using stable isotopes of $NO_3^-$, a shifting pattern of $NO_3^-$ sources from atmospheric deposition to inorganic N fertilizers in events with higher and longer duration of rainfall. The stable isotopes of PON confirmed that plant material (oak detritus, grass clippings) were the primary sources of PON in stormwater runoff. Our results demonstrate that practices targeting both inorganic and organic N are needed to control N transport from residential catchments to receiving waters.

## Introduction

Urbanization and anthropogenic activities have accelerated nutrient enrichment and water quality problems in urban waters [1, 2]. Nitrogen (N) is often a limiting nutrient in coastal

no grant number. The funders had no role in study design, data collection and analysis, decision to publish, or preparation of the manuscript.

**Competing interests:** The authors have declared that no competing interests exist.

waters [3–5], where excess N loading can lead to cultural eutrophication and algal proliferation [6]. Stormwater runoff is one transport vector of N from urban areas to receiving water bodies and thus a critical source to consider in finding strategies to reduce N enrichment of urban surface waters [7].

Nitrogen loading to urban waters via stormwater occurs in several forms, including inorganic forms i.e., ammonium ($NH_4^+$), nitrite ($NO_2^-$), and nitrate ($NO_3^-$) or organic forms i.e., dissolved organic N (DON) and particulate organic N (PON). Recent studies have shown that organic N can be a large proportion of N in urban stormwater, streams, estuaries and that portions of the DON pool can be bioavailable to the organisms that cause harmful algal blooms [6, 8–10]. The potential high proportion and bioavailability of DON imply the need to shift from traditional stormwater management practices that focus on inorganic N to the development of new strategies focusing on DON [9, 11]. Thus, we need a better understanding of the composition of N in urban waters, including information on the contribution of organic N in stormwater runoff. To date, only a handful of studies have investigated DON in urban stormwater runoff, and even fewer studies have considered PON in stormwater [10, 11].

The transport of N—both organic and inorganic—via urban stormwater runoff may be altered by changes in hydrology in watersheds [12, 13]. Changes in hydrology can be related to weather (e.g., rainfall, temperature) and/or land use changes (e.g., urbanization, dam and reservoir release), which influence the hydrological pathways [14–16]. Investigation of hydrologic trends and variability of N forms is useful to determine hydrologic variables that play important roles in N transport and to elucidate potential influences of rainfall patterns on N forms-specific transport mechanisms. Rainfall variables such as rainfall amount, duration, intensity, and antecedent dry periods have been used to determine the relationship between rainfall variables to nutrient and pollutant transport [17–19]. For example, a study by Schiff et al. [20] on stormwater runoff from parking lots in California, US showed that 18 measured compounds were inversely correlated to rainfall duration where longer rain events decreased the concentrations of the constituents in parking lot runoff. Liu et al. [21] found that TN concentration in runoff waters from an N-fertilized field was significantly correlated with rainfall amount as more rainfall generated more N runoff. Antecedent dry period was also reported to be a factor affecting nutrient transport in urban land where greater $NH_4$–N concentrations were observed after a long dry season [22].

In addition to considering all N forms, including organic forms, and relating their transport to rainfall variables, it is also important to elucidate sources of N to watersheds. Sources of N in urban landscapes include atmospheric deposition, anthropogenic (e.g., fertilizers, automotive detergent, pet waste), and organic materials (e.g., throughfall from the urban tree canopy, leaf litter, grass clippings) [12, 13, 23, 24]. A study in Pittsburgh, US, showed that $NO_3$–N sources in an urban stream included atmospheric deposition (6 to 34%), sewage (72 to 94%), and denitrification processes (7 to 22%) [24]. A study by Hobbie et al. [25] on stormwater runoff in St. Paul, Minnesota, US showed that N sources in the catchment included atmospheric deposition (19 to 34%), chemical fertilizers (37 to 59%), and pet waste (28%). Studies conducted by our group in Tampa, Florida, US revealed that atmospheric deposition was the major contributor of $NO_3$–N in residential urban catchments (35 to 71%), followed by chemical fertilizers (1 to 49%), and soil and organic materials (7 to 33%) [26, 27].

While numerous studies have investigated the sources of $NO_3^-$ in urban waters, very few studies have investigated the sources of organic N forms (DON and PON). In several urban landscapes, particulate matter and organic detritus are considered the main contributors to nutrient enrichment in stormwater runoff [11, 28, 29]. Bratt et al. [11] investigated stormwater outflow in a residential area in Minnesota, US, and suggested that leaf litter was a potential source of N via transport of street PON during high rainfall events and due to the

decomposition of litter by microbes. Janke et al. [28] in Minneapolis, US, across 19 urban watersheds demonstrated that TN was strongly correlated to street trees canopy (r = 0.68, $p<0.05$), suggesting that tree litter (e.g., leaves, seeds, flowers) contributes to N loading in stormwater. The study also showed that organic N concentration, which was the primary N form (mean: 71% of TN across all sites), was strongly correlated with street canopy (r = 0.71, $p<0.001$). For source tracking of PON sources, dual isotopes of $^{13}$C and $^{15}$N have been reported as a useful tool [30–32].

To aid in developing management strategies for achieving TN reduction in stormwater run-off, the objectives of this study were to investigate (1) all N forms, including DON and PON, in urban runoff, (2) the influence of rainfall variables (e.g., total rainfall, intensity, duration, antecedent dry period) on inorganic and organic N loads to runoff, and (3) the sources of N in stormwater runoff. Specifically, this study is one of the first to look at all N forms, including organic N, and include a source tracking component for PON.

## Materials and methods

### Site description

The study site was located in Lakewood Ranch, a residential neighborhood in Manatee County, Florida, US, which is a planned community with a population of 29,411 in 2018 (67% increase over the population of 19,816 in 2010). Total land area in the community is 6,153 ha, with 35 ha of water features that include 321 stormwater retention ponds. Surface runoff from residential areas enters into the stormwater ponds, which then flows to the Braden River, a sub-basin of the Manatee River, and a part of the Tampa Bay estuary watershed. Within the community, a study catchment of 3.89 hectares was identified, which consisted of 57.1% impervious surfaces (i.e., houses and roads) and 42.9% pervious surfaces (i.e., tree canopy and turfgrass). Surface runoff from this catchment enters into street gutters, and then into the stormwater drains before draining into a stormwater pond of 0.64 ha (Fig 1 and S1 Table). Aside from the stormwater retention pond, no other runoff control structures are present in the catchment. Homes within the catchment have downspouts that convey roof runoff to the lots. Soils are predominantly sandy spodosols with rapid infiltration rates. Like other counties in Florida, Manatee county (our study site) is subjected to a rainy season N fertilizer ban (Ordinance 11–20) adopted and enacted by the Board of County Commissioners of Manatee County, Florida. This ban prohibits the application of N fertilizers during summer months from June 1 to September 30. The annual precipitation, obtained from the Florida Automated Weather Network (FAWN), was 1285 mm (S1B Fig) and the annual mean temperature was 26.7°C.

### Study site instrumentation

We have several years of collaboration with the Homeowners Association (HOA) of the Lakewood Ranch, Bradenton, Florida. We were granted permission by the HOA to access the neighborhood and out study catchment to conduct research. As this is a residential community, no permits were required for our field work. Field instruments (laser flowmeter, rain gauge, and autosampler) were installed at the end of the stormwater pipe (76 cm diameter) that delivers runoff from the catchment to the stormwater pond (Fig 1). A laser flowmeter (Model 2160, LaserFlow Non-contact Velocity Sensor, Teledyne ISCO Inc., Nebraska, US) was used to measure flow, and a rain gauge was used to measure rainfall (Model 674, Teledyne ISCO Inc., Nebraska, US). The laser flowmeter has an accuracy of ±0.006 m at <0.3 m level change and ±0.012 m at >0.3 m level change. A refrigerated autosampler was used to collect runoff water samples (Avalanche, Teledyne ISCO Inc., Nebraska, US). The autosampler was

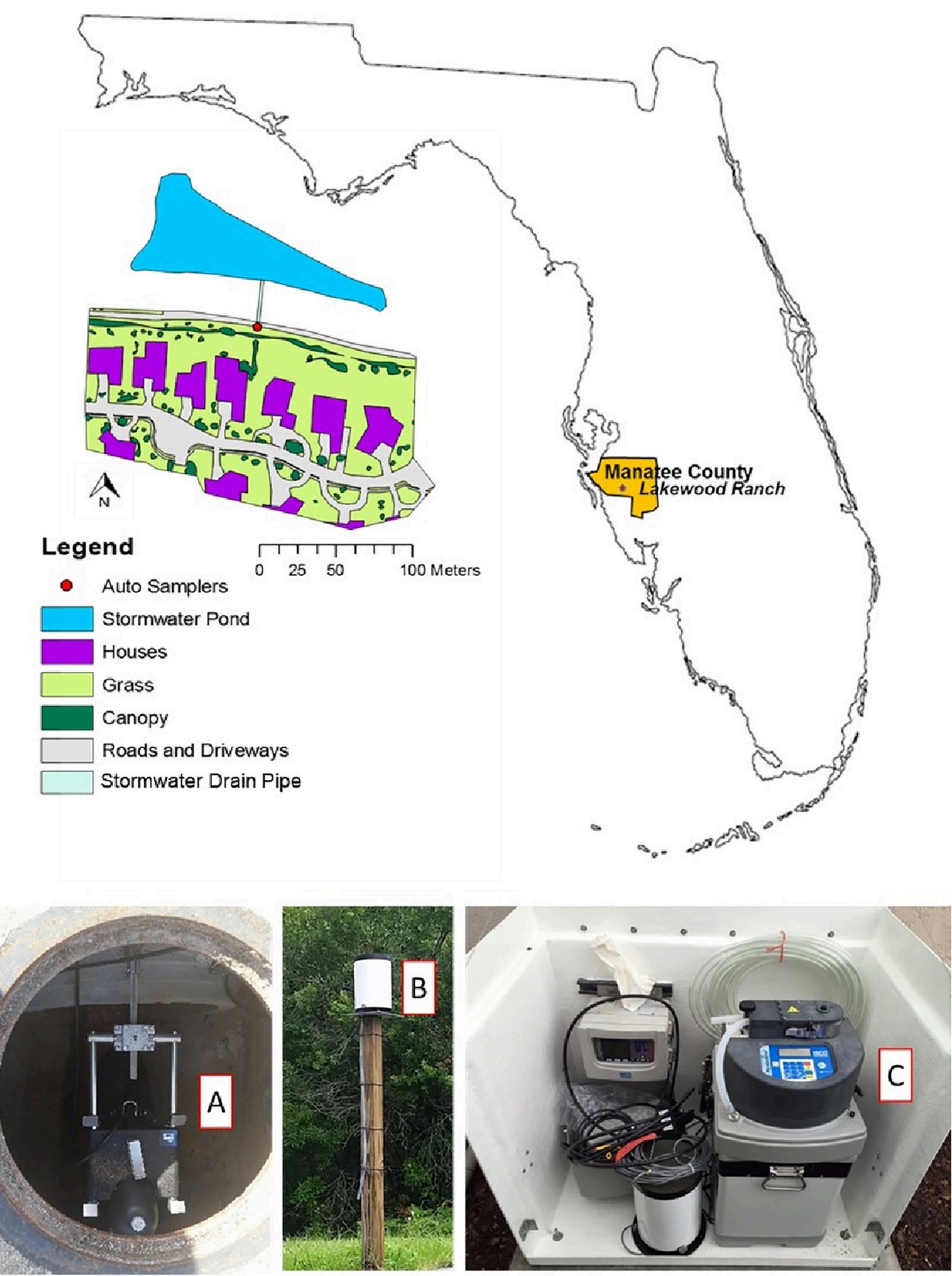

**Fig 1. Location map of the study site and residential catchment and various instruments installed in the stormwater outlet pipe, including A) laser flow meter, B) rain gauge, and C) autosampler.** The location map of the catchment was made using open data source, freely available at http:// geodata.myflorida.com/datasets/swfwmd::florida-counties in **ArcGIS 10.3.1 version.**

triggered to collect runoff samples when the rain gauge recorded 2.54 mm of rainfall in 10 minutes and/or the minimum level of water depth in the pipe (detected by the flowmeter) was 19.05 mm for 5 minutes. These criteria allowed the collection of runoff samples over several storm events of various duration and magnitude. As it is an urban residential site, there was no baseflow in the stormwater pipe. The autosampler contained 14 plastic bottles of 950 ml each and kept the samples stored at 4˚C. The flow meter was connected to a modem, which provided an online download of all data (rainfall amount and duration, runoff volume, and sample collection times) using the FlowLink 5.1 software (Teledyne ISCO Inc., Nebraska, US). The online connectivity system enabled monitoring and management of the sampling program remotely, using the FlowLink 5.1 software.

## Sample collection and preparation

Sampling was conducted in 2016 to capture the summer rainy season (June to September) and a month (May) preceding it. The plastic bottles from the autosampler were replaced at the end of each storm event that met the sampling threshold (2.54 mm of rainfall for 10 minutes and/or minimum water depth of 19.05 mm for 5 minutes in the pipe) or when all 14 bottles were full. The collected samples were transported on ice to the laboratory within 24 h of collection. Prior to the sampling, all bottles were acid-washed (10% hydrochloric acid, HCl) and rinsed with deionized water. The rainfall samples were collected from the sample bottle connected to the rain gauge.

## Nitrogen concentrations analysis

A subsample of the stormwater samples (n = 218) was vacuum-filtered using 0.45 μm glass fiber filters (GF/F) (Pall Corporation, Ann Arbor, MI) within 24 h of collection and analyzed using an auto-analyzer (AA3, Seal Analytical Inc., Mequon, WI) for $NH_4$–N and $NO_x$–N ($NO_3$–N + $NO_2$–N) with USEPA Method 350.1 [33] and 353.2 [34], respectively. The filtered (0.45 μm) and unfiltered samples were analyzed for total dissolved N (TDN) and TN, respectively, using oxidative combustion-chemiluminescence by Total Organic Carbon Analyzer (TOC-L CPH/CPN, Shimadzu Corp., Columbia, MD). Other N forms were calculated as follows: PON = TN–TDN; DON = TDN–($NO_x$–N + $NH_4$–N).

Flow-weighted mean concentrations (FWMC) in mg $L^{-1}$ were used to quantify the weighted concentration proportional to corresponding flow volume. The FWMC for each parameter are derived from the concentrations and flow volume for each sample during a specified window of time. This equation allows the concentration in each sample to be considered in light of the time and associated flow volume [35, 36]. The equation is as follows:

$$FWMC = \sum_1^n (ci * ti * qi) / \sum_1^n (ti * qi) \qquad (2-1)$$

where $c_i$ = concentration in the i$^{th}$ sample

$t_i$ = time (min) window of the i$^{th}$ sample

$q_i$ = flow volume in the i$^{th}$ sample

## Water isotopes analysis

Subsamples of filtered stormwater were refrigerated at 4˚C for stable isotopes of water ($H_2O$), i.e. oxygen ($\delta^{18}O$–$H_2O$) and hydrogen ($\delta D$–$H_2O$). A total of 10 rainfall samples over 10 storm

events and 176 runoff samples (additional 14 samples from event 1 and 7 samples from event 2 were omitted due to technical error) were analyzed for the isotopic composition of $H_2O$ using Chlortetracycline Liquid Chromatography Prep and Load (CTC LC-PAL) autosampler that was coupled with an off-axis integrated cavity output spectroscopy (OA-ICOS) water isotope analyzer (LWIA, Los Gatos Research, Mountain View, CA). A detailed description of $H_2O$ stable isotope analysis can be found in Lis et al. [37]. All stable isotopes values were reported as per mil (‰) according to Vienna Standard Mean Ocean Water (VSMOW) standards for O, and deuterium (D) with δ (‰) = 1000 x [($R_{sample}/R_{standard}$)]− 1, where $R$ represents the measured isotopic ratios of $^{18}O/^{16}O$, and D/H, respectively.

## Nitrate isotopes analysis

The filtered stormwater samples were frozen for $NO_3^-$ isotopic analysis, i.e. $δ^{18}O–NO_3^-$ and $δ^{15}N–NO_3^-$. Out of 22 storm events, runoff samples from 12 events (n = 176), and 12 rainfall samples were analyzed for the isotopic composition of $NO_3^-$. The 176 runoff samples for the isotopic composition of $NO_3^-$ were chosen based on the storm events that had more than 5 samples. Out of the 176 samples, 148 samples were suitable for $NO_3^-$ isotopic analysis as the rest of the samples had $NO_3^-$ concentrations below the detection limit. Analysis of $NO_3^-$ isotopes was conducted using the $AgNO_3$ method as described by [38]. All stable isotopes values are reported as per mil (‰) according to VSMOW standards for O and N with δ (‰) = 1000 x [($R_{sample}/R_{standard}$)]− 1, where $R$ represents the measured isotopic ratios of $^{18}O/^{16}O$, and $^{15}N/^{14}N$, respectively.

A Bayesian stable isotope mixing model Stable Isotope Analysis in R (MixSIAR) was used to quantify the contribution of $NO_3^-$ sources as described elsewhere [27, 39] Measured stable isotope data of our samples was compared with end- member values of $NO_3^-$ sources ($δ^{18}O–NO_3^-$ and $δ^{15}N–NO_3^-$) obtained from the literature [27, 40–42] to infer the $NO_3^-$ sources in our stormwater runoff. Potential end- members considered here included atmospheric deposition, $NH_4^+$ fertilizer, $NO_3^-$ fertilizer, nitrification, and soil and organic N in stormwater runoff (S2 Table). In the Bayesian mixing model, measured $δ^{18}O–NO_3^-$ and $δ^{15}N–NO_3^-$ values for each of the runoff samples from May to September 2016 (n = 148) were assigned as "customer" and the mean isotopic values of the $NO_3^-$ sources from the literature were assigned as "sources".

## Particulate organic nitrogen sources analysis

Isotopic characterization of $^{13}C$ and $^{15}N$ in stormwater particulates was used to investigate the potential sources of PON, based on methods similar to Kendall et al. [43]. Stormwater runoff samples were filtered through weighted 0.45 μm filters (GF/F). The retained particulates on the filter paper were oven-dried at ~80˚C. Potential sources of organic N from the residential catchment, i.e. grass clippings, oak leaves, and acorns, were collected in May 2016. These were washed with deionized water and oven-dried at ~80 °C before grinding to a fine powder. Both filter paper and the powder were then analyzed for $^{13}C$ and $^{15}N$, total carbon (C), and TN with an elemental analyzer (Costech ECS 4010 Elemental Combustion System) coupled to a mass spectrometer (Thermo Finnigan DeltaPlus XL, San Jose, California). All stable isotopes values are reported as per mil (‰) according to Vienna Pee Dee Belemnite (VPDB) standards for C, and atmospheric $N_2$ for N with δ (‰) = 1000 x [($R_{sample}/R_{standard}$)]− 1, where $R$ represents the measured isotopic ratios of $^{13}C/^{12}C$ and $^{15}N/^{14}N$, respectively.

The isotopic mixing model, IsoError, as described by Phillips et al. [44] was used to investigate the contribution of each potential source to runoff PON. The IsoError model is freely available from the US Environmental Protection Agency (www.epa.gov/eco-research/stable-isotope-mixing-models-estimating-source-proportions). The potential sources/end members $^{13}C$ and $^{15}N$ values are presented in the S3 Table.

## Statistical analysis

The SAS JMP Pro 13 software was used for statistical analysis in this study. Differences in the mean for response variables (i.e., TN, DON, PON, $NH_4$–N, $NO_x$–N, $\delta^{15}N$–$NO_3^-$, and $\delta^{18}O$–$NO_3^-$) and rainfall variables (i.e., antecedent dry period, amount, intensity and duration of rainfall), were input into a Pearson correlation to test for relationships among the variables. An alpha value equal to 0.05 was used as a threshold for statistical significance.

## Results and discussion

### Relationship between rainfall and stormwater runoff

Total rainfall during the study period (May to September 2016) was 835 mm (monthly mean: 167±54 mm), with the highest amount in August (254 mm) and the lowest in June (125 mm). Ten-year records (2006–2015) obtained from the FAWN show that total rainfall from May to September ranged from 522 mm to 1148 mm, with a mean value of 808±196 mm (S1 Fig).

From May to September 2016, 22 out of 75 storm events met the threshold requirement of sampling (2.54 mm of rainfall for 10 minutes and/or minimum water depth level of 19.05 mm for 5 minutes the in the pipe). The rainfall associated with the 22 events was 283 mm or 34% of total rainfall throughout the study period (835 mm), which included many small storms (<2.54 mm rain in 5 minutes) that did not meet the sampling threshold. In other words, a total of 3668 $m^3$ $ha^{-1}$ of rainfall was received in 22 storm events, with a range of 25.4 $m^3$ $ha^{-1}$ in event 4 to 1466 $m^3$ $ha^{-1}$ in event 1 (Fig 2A). The rainfall intensity associated with 22 storm events ranged from 1.8 to 32.4 mm $hr^{-1}$ (Fig 2B). The total flow associated with 22 sampled storm events varied with the rainfall characteristics, with the lowest of 1.5 $m^3$ $ha^{-1}$ in event 4 and highest of 1050 $m^3$ $ha^{-1}$ in event (Fig 2C).

S2A Fig displays percent runoff (fraction of rainfall that became runoff) during 75 storm events, including runoff associated with 22 sampled storm events. The frequency distribution graph shows that the percent runoff in 75 events ranged from 10 to 80% and that 22 storm events represented samples captured over this range of percent runoff, with 6 events with 10% runoff, 3 events with 20% runoff, 4 events with 30% runoff, 3 events with 40% runoff, 2 events with 50% runoff, 3 events with 70% runoff, and 1 event with 80% runoff (S2B Fig). The amount of rainfall was positively correlated with the amount of runoff ($r$ = 0.92 to 0.99, $p$<0.001), with slopes of 0.70 to 0.71, suggesting that 70 to 71% of total rainfall was converted to runoff (S3C Fig). Similar findings regarding large rainfall events and stormwater runoff were reported by Brezonik and Stadelmann [45].

### Concentrations and loads of nitrogen forms in stormwater runoff

**Overview of nitrogen forms.** Concentrations of TN in 22 storm events ranged from 0.28 to 10.11 mg $L^{-1}$, with FWMC of 2.45±0.2 mg $L^{-1}$ (S3 Fig). Among N forms, DON and PON had the highest concentrations in runoff samples. The FWMC of DON over 22 events ranged from 0.1 to 9.2 mg $L^{-1}$ (grand mean: 1.2 mg $L^{-1}$), whereas PON ranged from 0.07 to 8.3 mg $L^{-1}$ (grand mean = 0.9 mg $L^{-1}$).

Over the study period, the stormwater runoff transported 3.9 kg $ha^{-1}$ of TN, with DON as the dominant contributor at 72% (2.8 kg $ha^{-1}$), followed by PON at 10% (0.39 kg $ha^{-1}$), $NH_4$–N at 10% (0.39 kg $ha^{-1}$), and $NO_x$–N at 8% (0.31 kg $ha^{-1}$) (Fig 3A). Organic N forms (DON + PON) accounted for an average of 82% (range: 3 to 95%) of TN loads over the study period (Fig 3B). Our results are similar to other studies that observed ON as the most dominant form in stormwater samples [7, 10, 28]. For example, Lusk et al. [46] studied rainy season TN in stormwater runoff in a different Florida residential neighborhood and found that average TN

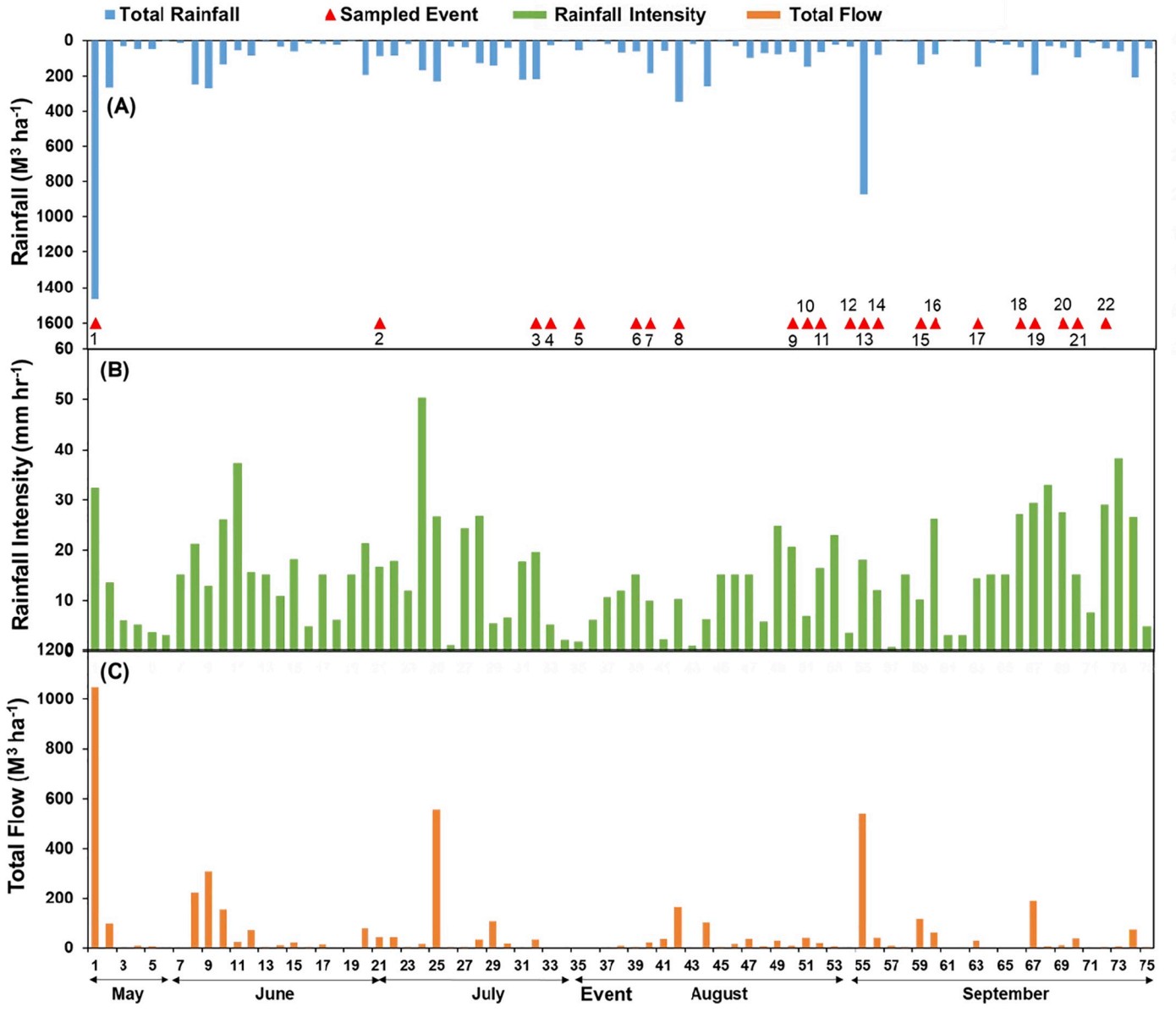

**Fig 2.** (A) Rainfall amount and information about 22 sampled storm events (red triangles), (B) rainfall intensity, and (C) stormwater flow associated with storm events from May to September, 2016.

over the season was 1.61 mg L$^{-1}$, of which DON was 37%, and PON was 25%. Other studies in Australia, Maryland, and Minnesota have reported stormwater ON proportion of 66%, and 52%, and 73%, respectively [7, 10].

**Inorganic nitrogen forms.** Dissolved inorganic N (DIN) forms had lowest concentration in runoff samples with FWMC NO$_x$–N ranging from 0.02 to 0.6 mg L$^{-1}$ (grand mean: 0.2 mg L$^{-1}$) and NH$_4$–N ranging from 0.1 to 0.5 mg L$^{-1}$ (grand mean: 0.2 mg L$^{-1}$) (S4 Table). The relative contribution of DIN was higher at the beginning of the wet season, especially during storm events with high rainfall and greater percent runoff (> 40%) (S2 and S3 Figs). For example, the prolonged high runoff in events 1, 8, and 13 resulted in higher DIN loads as compared to other shorter duration storms (Fig 3B). As such, both DIN forms were significantly

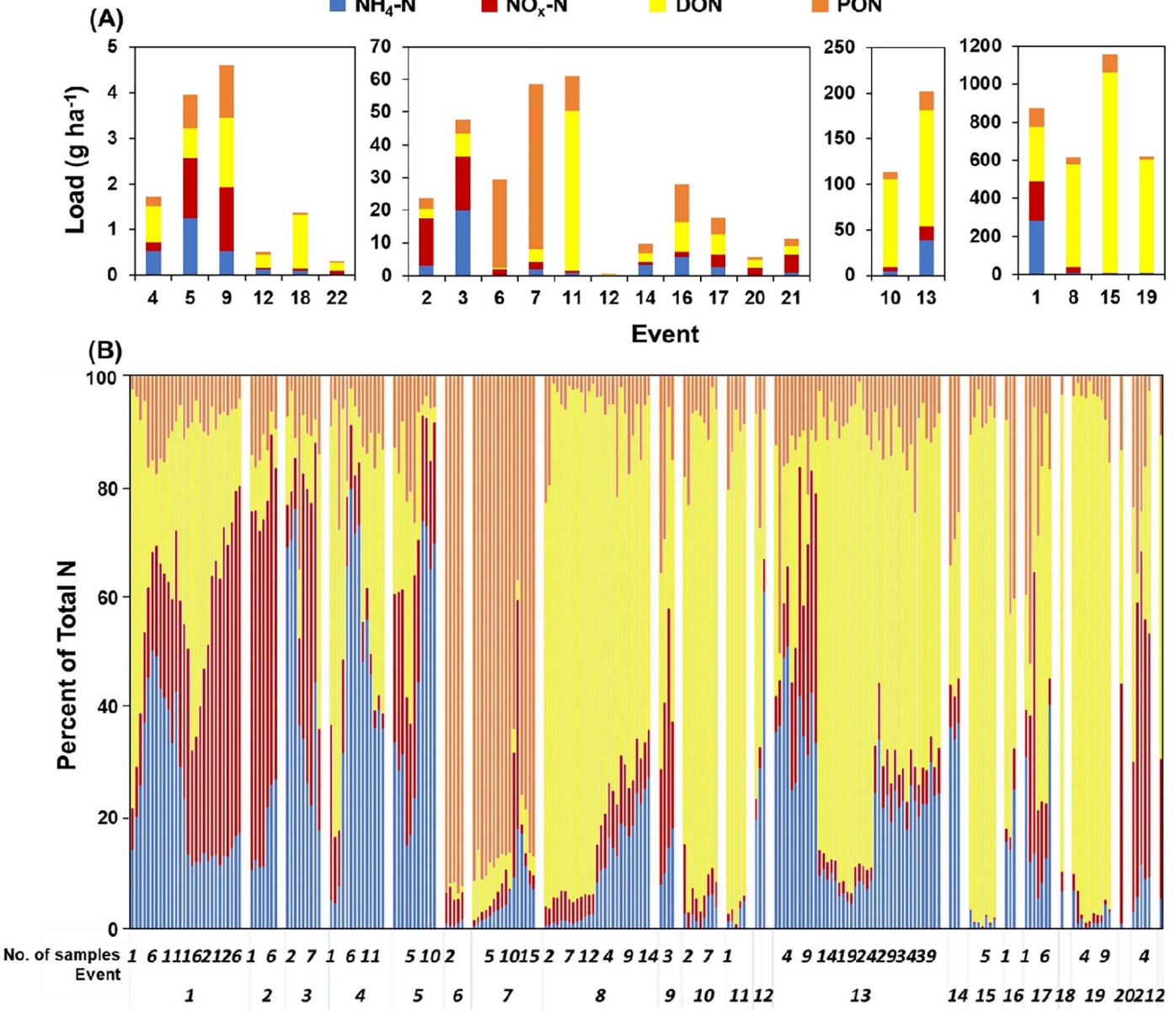

**Fig 3.** (A) Loads of various nitrogen forms separated into four loading groups and (B) percentage of nitrogen forms in 22 storm events from May to September 2016.

($p<0.05$) positive correlated with the duration of rainfall (S5 Table). Further, inorganic N forms were significantly correlated with total rainfall ($r = 0.90$, $p<0.001$), and antecedent dry period ($r = 0.94$, $p<0.001$), suggesting that a longer dry period followed by high rainfall events, such as the first event of the season, resulted in increased DIN loss to the runoff. In previous studies, pollutant build-up and wash-off have been observed [46]. For example, Li et al. [47] showed that the first flush effects in an urban catchment in China were driven by antecedent periods and rainfall amounts. Lewis and Grimm [22] observed a greater concentration of $NH_4-N$ in their urban catchment due to the longer antecedent dry period. Dry season or antecedent dry period provides an extended window for accumulation of dry deposits and nutrient build-up on urban surface such as the streets and rooftops before the onset of summer storms

[48, 49]. When storm events occur, these accumulated pollutants are washed off into storm-water runoff [50]. Kojima et al. [48] concluded that surface deposits such as road dust were the dominant contributor of $NO_x$–N in their urban catchment located in Chiba City, Japan.

In 22 storm events, the concentration of all N forms decreased after the onset of rainfall as runoff volume increased (S3 Fig). Further, when rainfall decreased and eventually stopped, runoff volume slowly decreased, and TN concentrations slightly increased, suggesting that the decrease in N forms in storm events was due to the dilution effect with rainfall. This trend is similar to a study by Miguntanna et al. [50], who also observed a decrease in TN concentration with an increase in rainfall duration.

**Organic nitrogen forms.**   The composition of DON over 22 storm events was relatively consistent, as DON was the most dominant form during the study period (S4 Table). Among rainfall variables, DON load was significantly correlated with only intensity ($r = 0.50$, $p < 0.05$), indicating that other variables such as rainfall amount, duration, runoff volume, and anteced-ent dry period did not significantly influence the relatively high contribution of DON to stormwater runoff. This suggests that a significant amount of DON can be transported in low and high rainfall events. In previous studies, DON sources were linked to organic fertilizers, soil organic matter, atmospheric deposition, and degradation of plant debris and leaf litters from urban landscapes [3, 9]. Hagedorn et al. [51] showed that there was an increase in DON export in the summer as a result of high decomposer activity and availability of fresh leaf litter. Decomposition of leaf litter has been reported to be one of the main contributors to DON in urban runoff [11, 52] suggesting that in order to reduce DON input in stormwater runoff, a control measure (e.g., street cleaning) prior to storm events could be used to eliminate the potential of PON decomposition to DON [29]. Selbig [29] showed that street cleaning and removal of leaf litter from street surfaces reduced the TN by 74% and TDN by 71% as this reduced the potential of N leaching from accumulated particulates. They further suggested that coherence, constancy, and timing of street cleaning and leaf removal are important factors to be measured in constructing effective stormwater management practices. Hochmuth et al. [53] reported that stormwater has the potential to leach nutrients from plant debris instantly, thus removal of leaf litter and plant debris must be done as soon as possible prior to storm events. Furthermore, a number of researchers, including our previous work in Tampa Bay, Florida, US, have concluded that a portion of the DON may be bioavailable and thus can be a source of water quality impairment in urban waters [3, 8, 39, 54].

In urban landscapes, engineered headwaters flowing over impervious surfaces and storm gutters rapidly deliver dissolved organic matter and N during storm events and increase partic-ulate inputs into urban stream networks [55]. Research has demonstrated that organic detritus and particulates can be the main contributors of N input into urban stormwater [29]. In this study, PON load was significantly correlated with antecedent dry days ($r = 0.67$, $p < 0.001$), rain-fall amount ($r = 0.73$ $p < 0.001$), and duration of rainfall ($r = 0.55$ $p < 0.001$) (S5 Table) indicating more particulate accumulation during longer dry periods, and high rainfall and prolonged storm events transport more PON. Within our study catchment, we observed particulates such as plant debris and leaf litter trapped on the grates of storm gutters. Our runoff samples recorded unusual high PON (events 6 and 7) concentrations (>50% of TN) that led to high TN concentrations (S4 Table and Fig 3). The data showed that these events had low rain and runoff volume, and occurred after storms with low rain and runoff volume (<40% runoff) (Fig 3A). Wei et al. [56] suggested that particulate matter might be intercepted by coarse surfaces during low stormwater runoff from previous events, thus debris stuck in the storm drains was not flushed, resulting in high PON concentrations in the subsequent samples.

Given the contributions of DON and PON to stormwater in our study and the potential for DON to contribute bioavailable N in urban waters, we recommend efforts to incorporate

organic N into N loading budgets and in designs for more effective stormwater management to improve the quality of urban waters, as also suggested by other researchers [9, 57].

## Variation in water isotopes in rainfall and stormwater runoff

In urban residential areas, stormwater runoff can originate from multiple water sources such as rainfall, municipal water used for irrigation, reclaimed water, or wastewater leaks. Therefore, we used water isotopes to determine the origin of water in stormwater runoff samples. Stable isotopes of $\delta^{18}O-H_2O$ and $\delta D-H_2O$ are known as environmental isotopic tracers that allow inference of the hydrological processes and origin of water in the aquatic systems [58]. The values of $\delta^{18}O-H_2O$ and $\delta D-H_2O$ in our rainfall samples (n = 10) ranged from –6.1‰ to –0.4‰ (mean: –3.5‰) and –36.4‰ to 10.1‰ (mean: –16.1‰), respectively (S4 Fig). The $\delta^{18}O$ and $\delta D$ of rainfall $H_2O$ can be used as an indicator of weather conditions where lower values indicate higher precipitation amounts [59]. This observation was confirmed in our three high rainfall events (1, 13, and 19) that had lower $\delta^{18}O$ and $\delta D$ of $H_2O$ as compared to other events (S5 Fig).

The runoff samples (n = 176) had $\delta^{18}O-H_2O$ ranging from –6.42‰ to 1.63‰ (mean: –3.15‰) and $\delta D-H_2O$ ranging from –43.35‰ to 18.64‰ (mean: –14.27‰) (S4 and S5 Figs). Our isotopic data are similar to a study conducted in South Florida during the summer wet season with mean values of $\delta^{18}O-H_2O$ at –3.38‰ and $\delta D-H_2O$ at –16.5‰ [59]. The global meteoric water line (GMWL) is an equation interpreted as $\delta D-H_2O = 8\delta^{18}O-H_2O + 10$, which represents the relationship between H and O isotopes of water [60]. The isotopic composition of our runoff samples was identical with GMWL, as shown in S4 Fig with $\delta D-H_2O = 8\delta^{18}O-H_2O + 11$. The isotopic composition of runoff and rainfall samples at the study site was also similar (i.e., $\delta D-H_2O = 8\delta^{18}O-H_2O + 11$), indicating that all runoff in 22 storm events originated from local rainfall and not from other sources (e.g., municipal water, reclaimed water, wastewater).

Our water isotopes data showed differences for both $\delta^{18}O-H_2O$ and $\delta D-H_2O$ in rainfall and sequential runoff samples across individual events (S5 Fig). These differences and variations might be attributed by the effects of rainfall water being transmitted thorough canopy such as throughfall and stemflow before falling into the ground and emerging as surface runoff [61] and as condensation and evaporation processes that occurred as water was conveyed from land to stormwater network [27, 42].

## Variation in nitrate isotopes in rainfall and stormwater runoff

The $\delta^{15}N$ and $\delta^{18}O$ of $NO_3^-$ in rainfall samples over 12 storm events (n = 148) were –4.43‰ to 5.69‰ (mean: –5.30‰) and 36.70‰ to 67.08‰ (mean: 60.52‰), respectively (S6 Fig). The rainfall $\delta^{15}N-NO_3^-$ values were between the values reported by Felix et al. [40] and Buda and Dewalle [62], who reported $\delta^{15}N$ of –5.7‰ to 11.3‰ (mean: 3.2‰) and –0.6‰ to 5.0‰ (mean: –2.9‰), respectively. The $\delta^{18}O-NO_3^-$ were also similar to Felix et al. [40] and Buda and DeWalle [62] who reported values of 32.2‰ to 68.7‰ (mean: 47.9‰) and 11.9‰ to 70‰ (mean: 43.8‰), respectively. Studies on seasonal patterns showed that the isotope values of $\delta^{15}N-NO_3^-$ and $\delta^{18}O-NO_3^-$ in precipitation varied due to the differences in $NO_3^-$ sources and atmospheric oxidation pathways [40, 42, 62]. For example, Hastings et al. [63] reported that $\delta^{15}N-NO_3^-$ was lower in the cool season (October to March) compared to the warm season (April to September) in their studied region (Bermuda) due to more lightning during the warm season. Studies demonstrated that in the summer season, atmospheric reactions are dominated by oxidation of $NO_x$ through hydroxyl radicals (OH), which causes lower $\delta^{18}O-NO_3^-$, whereas in the winter season, the reaction between $NO_x$ and $O_3$ results in higher $\delta^{18}O-$

$NO_3$ [62, 63]. The $\delta^{15}N–NO_3^-$ of stormwater runoff samples ranged from –9.72‰ to 8.06‰ (mean: 1.02‰), whereas $\delta^{18}O–NO_3^-$ of stormwater runoff samples ranged from –9.19‰ to 59.70‰ (mean: 26.93‰) (S6 Fig). The $\delta^{15}N–NO_3^-$ in our sequential 10 minute collected run-off samples showed wide variation within the individual storm events. The differences in the intra-storm variation of $\delta^{15}N–NO_3^-$ ranged from 0.39‰ to 10.06‰ for all 12 storm events (mean: 5.04‰), with the highest variation of 10.06‰ in event 19 (S6A Fig). The $\delta^{18}O–NO_3^-$ in runoff samples also showed variation within individual storm events (S6B Fig). The intra-storm variation of $\delta^{18}O–NO_3^-$ ranged from 14.17 to 56.67‰ for all 12 storm events (mean: 33.28‰), with the highest difference of 56.67‰ in event 8. In summary, our results showed that the variation in the $\delta^{15}N–NO_3^-$ and $\delta^{18}O–NO_3^-$ in stormwater runoff samples might be attributed to the changes in the sources of atmospheric $NO_3^-$ and mixing of $NO_3^-$ from different sources in the catchment during storm events [26, 62, 64, 65].

## Changing sources of nitrate in storm events

The $\delta^{15}N$ and $\delta^{18}O$ of $NO_3^-$ in our runoff samples were in the range observed for multiple sources such as atmospheric deposition (that includes vehicle emission and lightning), nitrification, inorganic fertilizer ($NO_3^-$ and $NH_4^+$), and soil and organic N (S2 Table and Fig 4). The $\delta^{18}O–NO_3^-$ values from –10‰ to 10‰ have been suggested to be indicative of nitrification, which can be calculated using this formula: $\delta^{18}O–NO_3 = 1/3\ \delta^{18}O–O_2 + 2/3\ \delta^{18}O–H_2O$ [42]. The expected values of $\delta^{18}O–NO_3^-$ for nitrification in our samples, obtained from the equation, ranged from 3.55‰ to 8.92‰.

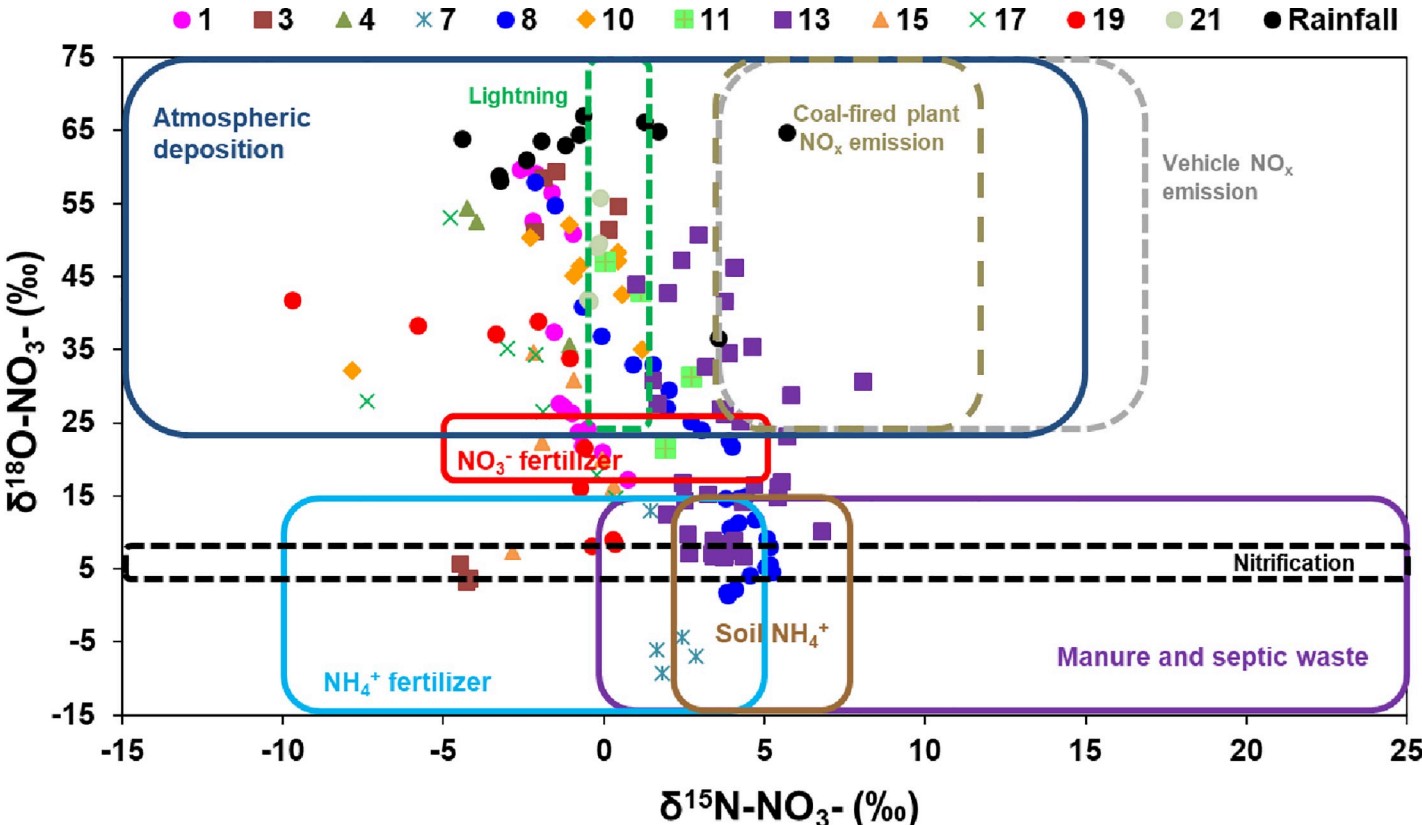

**Fig 4. $\delta^{15}N–NO_3^-$ and $\delta^{18}O–NO_3^-$ in 12 selected storm events from May to September 2016.** Boxes indicate the range of the $\delta^{15}N–NO_3^-$ and $\delta^{18}O–NO_3^-$ values for $NO_3^-$ sources according to Kendall et al. (2007), Heaton (1990), and Felix et al. (2015) as shown in S2 Table.

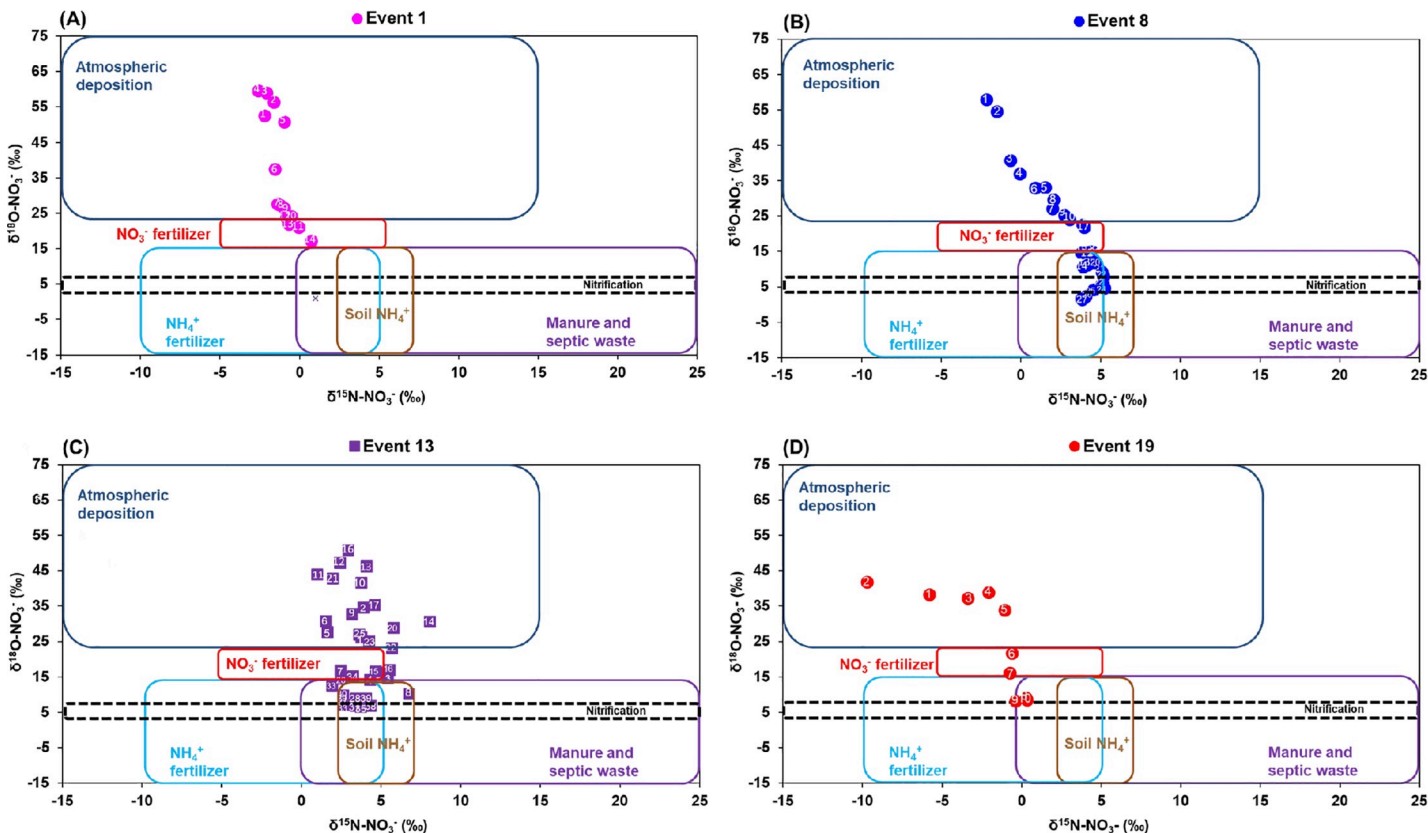

**Fig 5.** $\delta^{15}N–NO_3^-$ and $\delta^{18}O–NO_3^-$ in four individuals storm events (A) event 1 (number of samples, n = 14), (B) event 8 (n = 27), (C) event 13 (n = 39), and (D) event 19 (n = 10) from May to September, 2016.

We observed a shifting pattern of $NO_3^-$ sources in individuals storm events, especially in events with high and longer duration of rainfall such as events 1, 8, 13, and 19 (Fig 5). In these four events, atmospheric deposition was initially the main $NO_3^-$ source, however, as the storm events progressed, the main source of $NO_3^-$ changed to $NO_3^-$ fertilizer, eventually, including multiple other sources such as $NH_4^+$ fertilizers, nitrification, and soil and organic N. This pattern suggests that when the storm events started, the earliest runoff samples were from the direct atmospheric deposition. As the storm event progressed, the runoff began carrying other landscape sources of $NO_3^-$. This provides evidence that stormwater runoff can mobilize and transport inorganic N fertilizers from urban landscapes (Figs 4 and 5), and, therefore there is some validity to the claims that urban N fertilizers have the potential to be carried via stormwater runoff to receiving water bodies.

## Identifying nitrate sources in stormwater runoff

While nitrate concentrations were lower than organic N forms in our samples, the urban fertilizer ordinances in our study area necessitate some discussion of $NO_3^-$ sources—as the premise behind the regulatory fertilizer bans in several Florida counties is that summer rains mobilize bioavailable inorganic N to stormwater runoff. To date, however, no local studies have attempted to verify or provide data in support of this premise. We used a Bayesian mixed model to separate the contribution of various sources of $NO_3^-$. Consistent with our previous research in Tampa Bay region [26], we found that atmospheric deposition was an important

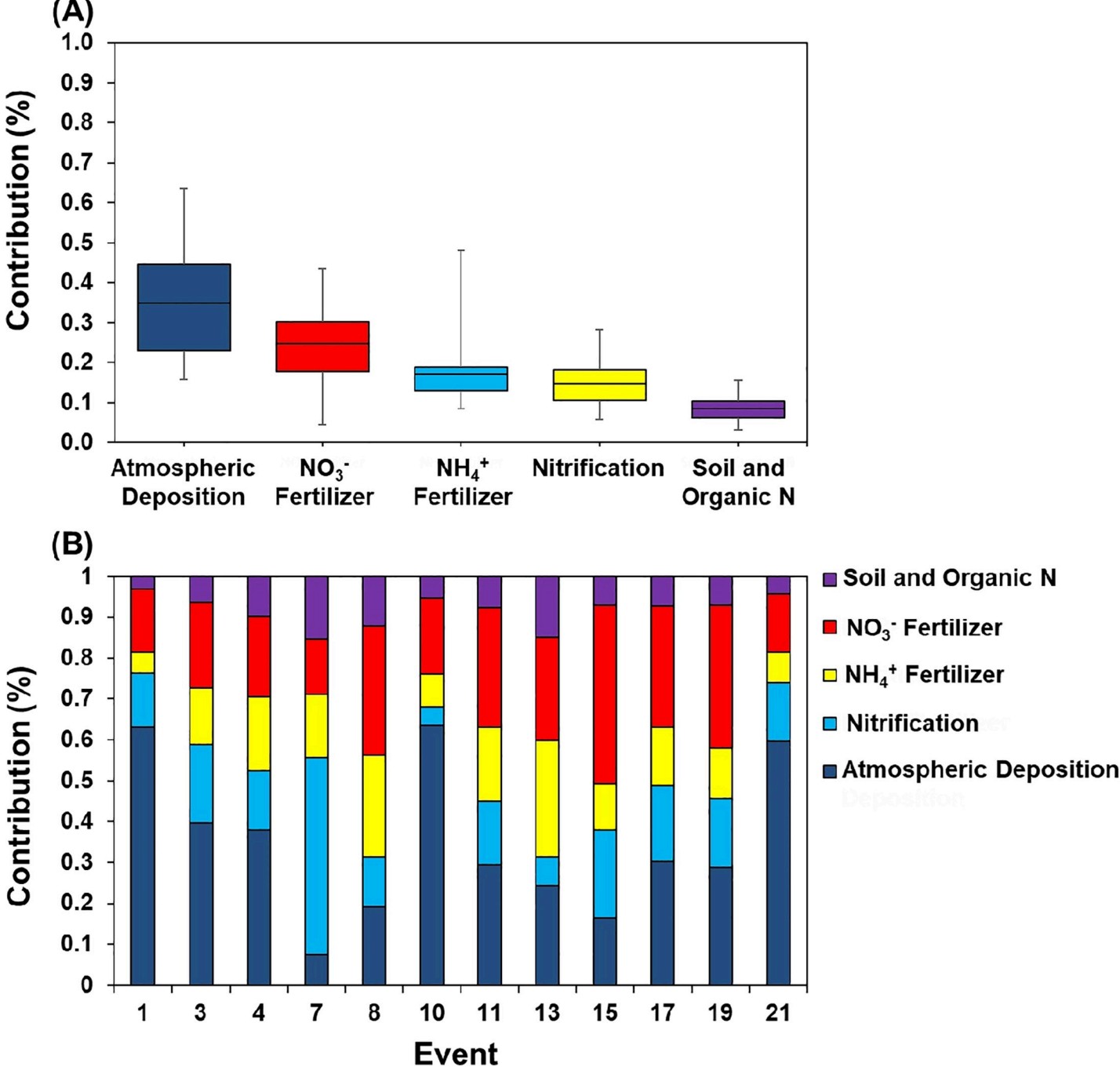

**Fig 6.** (A) Overall mean percent contribution of five $NO_3^-$ sources and (B) Mean percent contribution of $NO_3^-$ sources in 12 individual storm events from May to September, 2016.

contributor of $NO_3^-$ in stormwater runoff (34.9%), followed by $NO_3^-$ fertilizer (24.7%), $NH_4^+$ fertilizers (17.2%), nitrification (14.8%), and soil and organic N sources (8.4%) (Fig 6A). In this study, we applied the Bayesian mixing model to individual storm events to identify the $NO_3^-$ sources to capture source variability between storm events. Among 12 storm events, 7 events had isotopic signatures of $NO_3^-$ dominated by atmospheric deposition (mean: 30.1 to

63.5%), 3 events were dominated by $NO_3^-$ fertilizer (mean: 31.6 to 43.5%), 1 event (#13) was dominated by $NH_4^+$ fertilizers (48.1%), and another event (#7) was dominated by nitrification (28.3%) (Fig 6B). The changing sources contribution in different events highlights the variable nature of $NO_3^-$ sources in the landscape and the complex interplay of rainfall with landscape features on changes in the source contributions in different storm events.

Inorganic N fertilizers ($NO_3^-$ and $NH_4^+$) were the 2nd and 3rd largest contributors to $NO_3$–N in our runoff samples (Fig 6A). The Bayesian mixing model indicated that $NO_3^-$ fertilizer contributed 13.5% to 43.5% (mean: 24.7%), whereas $NH_4^+$ fertilizer contribution to $NO_3$–N was 4.6% to 48.1% (mean: 17.2%). Several studies on potential sources of anthropogenic N found that N input from fertilizer was the largest source of anthropogenic N fluxes from landscapes into the aquatic ecosystems [66–68]. Inorganic N fertilizers are commonly used in urban residential areas as part of landscape management to maintain plant quality. Inorganic N fertilizer was the dominant contributor in events 8, 15, and 19, which were the storm events with high rainfall amounts that resulted in high runoff flows. Even though our study was conducted during the summer season when application of N fertilizers is prohibited in this residential catchment from June 1 to September 30 (exception of event 1), the source of inorganic fertilizers might be from the residues of long-term controlled release fertilizers applied before the ban period (i.e., June 1).

## Sources of particulate organic nitrogen in stormwater runoff

The values of $\delta^{15}N$ and $\delta^{13}C$ in stormwater runoff particulates i.e. PON (n = 163 in 19 storm events) ranged from –1.99‰ to 6.27‰ (mean: –1.03±1.38‰) and –28.31‰ to –19.46‰ (mean: –23.04±1.73‰), respectively (Fig 7). These isotopic values appeared to be from a mixture of grass clippings of St. Augustine (*Stenotaphrum secundatum*) and acorns and leaves of live oak (*Quercus virginia*) trees. The mean $^{15}N$ for grass clippings, acorns, and oak leaves was –0.46 ‰ (range: –1.93 to 0.68), 1.58‰ (range: 1.55 to 1.60), and –1.24‰ (range: –1.70 to –0.83), respectively (Fig 7 and S3 Table). Whereas the mean $^{13}C$ for St. Augustine grass, acorns, and oak leaves was –14.2‰ (range: –17.8 to –11.6), –29.4‰, (range: –30.79 to –28.44) and –28.8‰ (range: –29.90 to –27.41), respectively. This data was modeled using the IsoError mixing model (Phillips et al., 2005), which estimated that acorns (41%), followed by grass clippings (32%), and oak leaves (27%) were the dominant contributor of PON in stormwater runoff (Fig 7).

Acorns drop from live oak trees in Florida during October each year, which then decompose on the ground for the following several months. Live oak trees naturally shed old leaves in spring (February to March) as the new leaves emerge, which then slowly decompose over the dry season. When the wet season begins in June, these partly decomposed materials (acorns and oak leaves) are carried by the stormwater runoff into the gutter and then contribute PON in runoff [69]. This hypothesis is supported by several studies that demonstrated the decomposition of leaf litter as a contributor to dissolved N in stormwater [11, 28, 29, 70]. These studies suggested particulates were decomposed by vehicle activity on the road surface, movement during storm events, and further decomposition in road gutters, thus contributing PON while DON was gained from leaching of freshly fallen litter. In this study, the particulates from oak detritus (acorn and oak leaves) accounted for 59% of PON in stormwater samples.

Urbanization often leads to changes in vegetation from trees to grasses [71]. In our study catchment, turfgrass (St. Augustine) covered about 51% of the total area (S1 Table), thus making grass clippings one of the most abundant sources of PON. Newcomer et al. [72] suggested that grass clippings are a potential source of labile N that can be readily mineralized. Lusk et al. [73] showed that DON was the main N form in leachate from turfgrass (St. Augustine) and

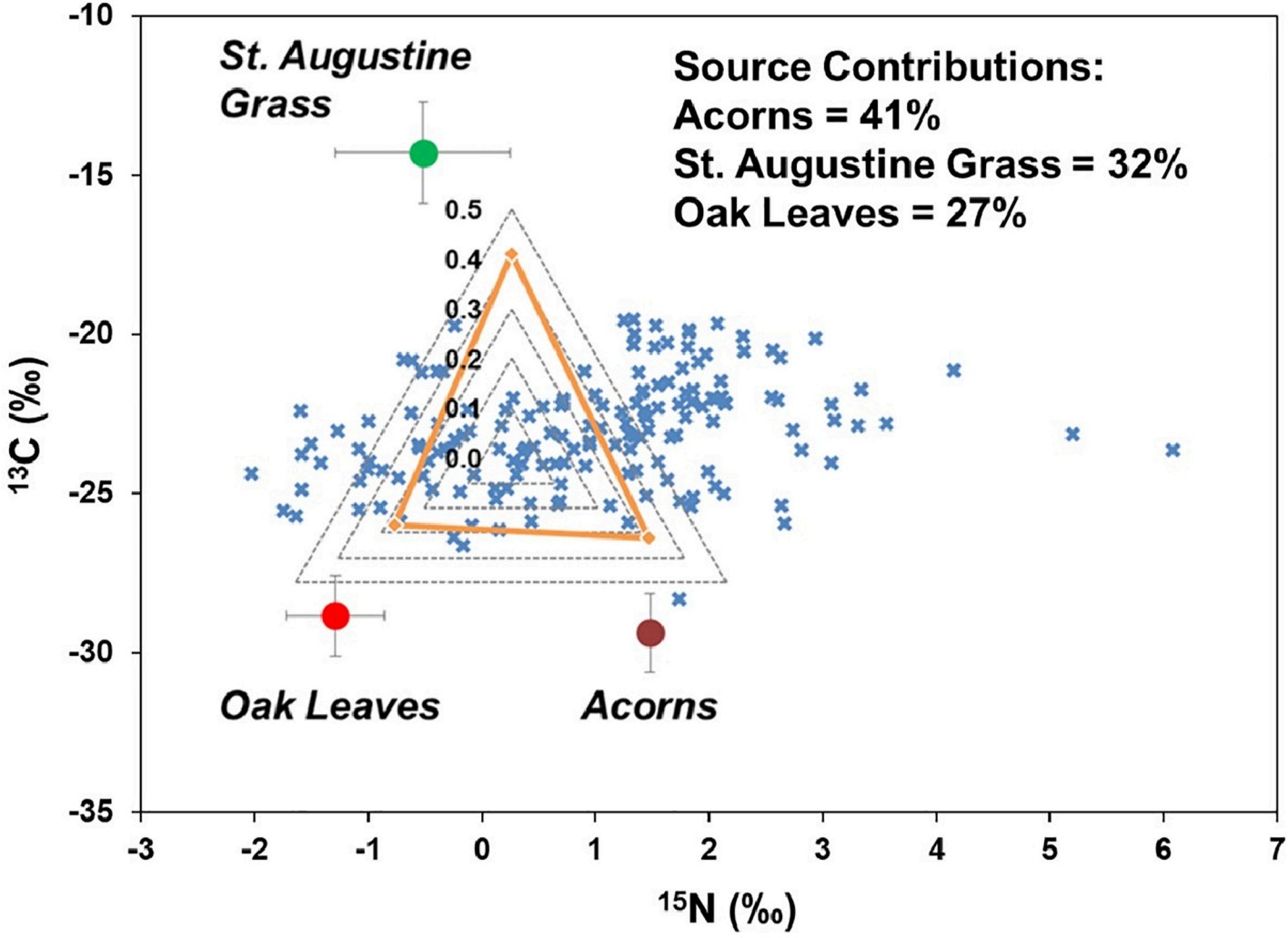

**Fig 7. Values of δ$^{15}$N and δ$^{13}$C for particulate organic N (PON) in stormwater runoff samples (blue crosses) and end-members (acorns, oak leaves, and St. Augustine grass).** Source proportions of three end-members were derived from the IsoError mixing model and are estimates of the proportion of each source to PON in the stormwater runoff samples.

suggested that root and microbial exudates in turf systems can convert inorganic fertilizer N to organic N that can be leached as DON in short periods of times (days to week).

## Conclusions

This study was conducted to investigate the composition of N forms and sources of $NO_3^-$ and PON in stormwater runoff. Among all N forms, DON was the dominant N form (mean: 47%) in stormwater runoff from May through September 2016 suggesting that management of runoff in terms of N should target not only inorganic N, but also organic N. Among rainfall variables, DON was positively correlated to only intensity, indicating that higher intensity of rain may be flushing out DON from soils and causing leaching of DON from particulates in the catchment. Statistical analysis showed that longer antecedent dry season and high rainfall amounts are more significant drivers for DIN transport to water bodies. Nitrate loading to stormwater runoff was derived from the mixing of multiple sources with atmospheric deposition as the dominant $NO_3$–N source (34.9%) followed by $NO_3^-$ fertilizer (24.7%), $NH_4^+$

fertilizer (17.2%), nitrification (14.8%), and soil and organic N (8.4%). The isotopic data showed a shifting pattern of $NO_3^-$ sources in events with high and longer duration of rainfall, suggesting that stormwater management to reduce N transport should include approaches that incorporate both rainfall and stormwater runoff in designs, such as stacking of best management practices including rain gardens, roof gardens, and permeable pavements. These low-impact development methods will not only reduce the momentum and erosive power of the stormwater and provide more time for the water to infiltrate into the ground, but also help to filter N before the stormwater enters into connected urban waters. Our data provides validity to the claims that inorganic fertilizers have the potential to runoff in urban residential areas. The sources of PON in stormwater runoff were acorns (41%), grass clippings (32%), and leaves from live oak trees (27%) present in the residential catchment. The decomposition of PON is a potential contributor to DON loading in urban runoff, suggesting an approach for N reduction should take place prior to storm events such as removal of organic materials (e.g., leaf litters, grass clipping, animal wastes) from urban pervious and impervious surfaces. Further, research on understanding the sources of DON, holistic evaluation of ways to prevent DON leaching, and the impacts of stormwater pond designs on N removal from residential catchments is needed.

## Supporting information

**S1 Fig.** (A) Annual and May to September rainfall from 2006 to 2016, (B) comparison of mean 10 years (2006–2015) and 2016 rainfall, and (C) comparison of monthly rainfall from May to September for 2006–2015 and 2016 in the study site located in Bradenton, Florida. Red bar in (A) indicates data of sampling year (2016).
(TIFF)

**S2 Fig.** (A) Percent runoff of total rainfall, (B) frequency distribution of percent runoff, and (C) relationship between rainfall and runoff amount for total 75 storm events and sampled 22 storm events from May to September, 2016 (line fits for both 75 and 22 events are shown).
(TIFF)

**S3 Fig. Flow-weighted mean concentrations of various nitrogen forms in 10-minute samples collected in 22 individual storm events from May to September, 2016.**
(TIFF)

**S4 Fig. Relationship between $\delta^{18}O–H_2O$ and $\delta D–H_2O$ for rainfall (n = 10) and runoff samples (n = 176) collected during 22 storm events from May to September, 2016.**
(TIFF)

**S5 Fig.** Variation in stable isotope composition (A) $\delta^{18}O–H_2O$ and (B) $\delta D–H_2O$ in rainfall (n = 10) and stormwater runoff samples (n = 176) collected during 22 storm events from May to September, 2016.
(TIFF)

**S6 Fig.** Variation in stable isotope composition (A) $\delta^{15}N\text{-}NO_3^-$ and (B) $\delta^{18}O\text{-}NO_3^-$ in rainfall (n = 12) and stormwater runoff samples collected during 22 storm events from May to September, 2016.
(TIFF)

**S1 Table. Pervious and impervious area of residential catchment located in Lakewood Ranch, Bradenton, Florida, United States.**
(PDF)

**S2 Table. End-member literature values of $\delta^{18}O–NO_3^-$ and $\delta^{15}N–NO_3^-$.**
(PDF)

**S3 Table. $^{13}C$ and $^{15}N$ of PON in various landscape sources and collected stormwater runoff samples.**
(PDF)

**S4 Table. Runoff volume variables, flow-weighted mean concentration of nitrogen forms, and $\delta^{18}O–NO_3^-$ and $\delta^{15}N–NO_3^-$ values for 22 storm events from May to September, 2016.**
(PDF)

**S5 Table. Pearson correlation among rainfall variables and nitrogen forms from May to September, 2016.**
(PDF)

## Acknowledgments

We thank Drs. Kati Migliaccio, Andrew Koeser, and John Thomas for serving on Jariani Jani's PhD advisory committee. The first author thanks Ministry of Higher Education, Malaysia for providing a PhD fellowship. We thank Stefan Kalev, former MS student, for his help and assistance in troubleshooting field instruments and collecting stormwater runoff samples. advisory committee. This project would not have been possible without tremendous support, cooperation, and advocacy from residents and homeowner association of Lakewood Ranch community, located in Manatee County, Florida.

## Author Contributions

**Conceptualization:** Gurpal S. Toor.

**Funding acquisition:** Gurpal S. Toor.

**Methodology:** Gurpal S. Toor.

**Project administration:** Gurpal S. Toor.

**Resources:** Gurpal S. Toor.

**Supervision:** Gurpal S. Toor.

**Writing – original draft:** Jariani Jani.

**Writing – review & editing:** Yun-Ya Yang, Mary G. Lusk, Gurpal S. Toor.

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
