## [Decision Letter · Decision Letter 0]

21 Oct 2019

PONE-D-19-21012

Composition of Nitrogen in Urban Residential Stormwater Runoff: Variation in Concentrations, Loads, and Source Characterization of Nitrate and Organic Nitrogen

PLOS ONE

Dear Dr. Toor,

Thank you for submitting your manuscript to PLOS ONE. After careful consideration, we feel that it has merit but does not fully meet PLOS ONE’s publication criteria as it currently stands. Therefore, we invite you to submit a revised version of the manuscript that addresses the points raised during the review process.

All three reviewers agree that the manuscript requires minor revisions. Please address all revisions as suggested by the reviewers.

We would appreciate receiving your revised manuscript by Dec 05 2019 11:59PM. To enhance the reproducibility of your results, we recommend that if applicable you deposit your laboratory protocols in protocols.io, where a protocol can be assigned its own identifier (DOI) such that it can be cited independently in the future. For instructions see: http://journals.plos.org/plosone/s/submission-guidelines#loc-laboratory-protocols

We look forward to receiving your revised manuscript.

Kind regards,

Julian Aherne

Academic Editor

PLOS ONE

Journal Requirements:

Additional Editor Comments (if provided):

Your manuscript requires minor revisions based on the comments from three reviewers (all three agree that the manuscript requires minor revisions). Please ensure that in your revised manuscript you address all comments (note: one reviewer has provided comments directly on a PDF copy of the manuscript). I looked forward to the revised manuscript.

Reviewers' comments:

Reviewer's Responses to Questions

**Comments to the Author**

1. Is the manuscript technically sound, and do the data support the conclusions?

Reviewer #1: Yes

Reviewer #2: Yes

Reviewer #3: Yes

2. Has the statistical analysis been performed appropriately and rigorously? 

Reviewer #1: Yes

Reviewer #2: Yes

Reviewer #3: Yes

3. Have the authors made all data underlying the findings in their manuscript fully available?

Reviewer #1: Yes

Reviewer #2: Yes

Reviewer #3: Yes

4. Is the manuscript presented in an intelligible fashion and written in standard English?

Reviewer #1: Yes

Reviewer #2: Yes

Reviewer #3: Yes

5. Review Comments to the Author

Reviewer #1: General observations

The manuscript represents a valuable addition to the literature on stormwater (SW) quality and general approaches to quality controls. While the initial approach to SW control focused on removals of suspended solids with attached chemicals, environmental concerns about nutrients exported with SW from urban catchments were addressed at a later date and often indicated transformations of various species of N and P in Best Management Practice facilities, or poor experimental methods and erroneous measurements. Hence, scientifically rigorous contributions, as the one reviewed herein, are useful precursors of further progress in this field. The manuscript is well thought out and written, the objectives clearly stated, experimental methods are advanced, data statistically analyzed, the conclusions well supported by the data presented, and in principle, the reviewer recommends its acceptance, pending some relatively minor corrections/clarifications improving the clarity of presentation.

Speaking of clarity, the reviewer is slightly concerned that the authors depart in some cases from a common usage of hydrological terms and units, when describing the catchment hydrology (defined as those appearing e.g., in the Journal of Hydrology). I realize that these simple formal changes may involve enough work, and therefore, leave it up to the Editor to decide whether those changes are worthwhile. At the same time, the usage in the manuscript is somewhat distracting.

Specific comments

Line (L) 66/67 – flow (presuming runoff flow) should not be listed as a rainfall variable.

L101 – runoff volume is not a rainfall variable

L107-123 – the study catchment description is incomplete; for applications elsewhere, one would be interested e.g. in an average lot size; whether the impervious areas are directly connected to sewers, or some of them discharge on pervious parts of the catchment; do roofs discharge onto lots, or are they connected to sewers?; soil type (and even just a qualitative description of soil infiltration); whether any runoff controls are used on individual lots, or in the catchment.

L116 – surface runoff does not enter into curbs, but street gutters and then into storm sewers (or drains) before draining…; any comments on the N fertilizer ban compliance?

L124 – annual precipitation and air temperatures, any other comments re the catchment climate.

L126 – the rain gauge is not installed in the storm sewer outlet, but at, or by; and incidentally, the one shown in Fig. 1 is most likely “shaded” by the nearby trees (bias in in rainfall capture) and measurement accuracy.

How were the rainfall events defined? They are usually defined by the minimum rainfall depth (e.g., > 2 mm) and minimum inter-event time (e.g., 6 hours) – needed to distinguish between events with shorter periods of no rain and two events generated by different whether systems.

L130-132 – flow monitoring – the model number?, some estimates of accuracy?

L163-167 – please check eq. (2-1) – I believe that it is incorrect (assuming that n is a sample count); as it is written, I can cancel “ti*ci” in the numerator and denominator and the equation then reads FWMC = sum (ci), from 1 to n.

L170 – rainfall samples – I do not recall reading about them earlier, how and where they were collected?

L221 – the rainfall variables are listed correctly (contrarily to Line 101); rainfall “amount” is generally measured as rainfall depth [mm]; the intensity – is that event mean intensity?

L226 and elsewhere – total rainfall of 1,055 mm (this applies throughout the manuscript)

L232 – 233 – two decimal points 2.54 mm, 19.05 mm is excessive and misleading as to measurement accuracy, use 2.5 and 19.1 mm; …2.5 mm of rainfall in 10 minutes…; …the in the…, L233, delete the first the.

L236 – the rainfall intensity …ranged from 2.0 to 5.1 mm hr-1 …. – the intensities were averaged over the storm event duration; for later discussion of N export from the catchment, peak intensities over some fixed time interval (close to the catchment time of concentration, perhaps 15 – 30 minutes, would be more relevant.

L237-238 – unusual units of total volume (rather than amount) of runoff – the larger number equals 554 mm of runoff and the smaller 168 mm of runoff (from the catchment studied). 1 litre is perhaps much too small unit of volume; why not m3 ?

L276-277 – it would be useful to define “high” rainfall and runoff quantitatively

L281-284 – this has been defined in the literature as “pollutant build-up and wash-off” (e.g., in the US EPA SWMM model, or see e.g. Vaze, J. and Chiew, H.S. (2003). Study of pollutant washoff from small impervious experimental plots. Wat.Res. Research, 39(6), 1160; doi: 10.1029/2002WR001786

L302 – rainfall amount listed twice

L383 - …winter season, reaction occurs….

L496 – 497 – it is not just the issue of age of sanitary and storm sewers, but more importantly, no cross-connections (misconnections) between both systems.

L521-523 – it is hard to visualize “rotting” leaves and acorns in street gutters in an obviously upscale residential area; does this really happen?

L542 – perhaps “longer antecedent dry season”

L556 – it is easy and “logical” to suggest cleaning streets before storm events, but that is a very costly proposition. Perhaps one should somehow temper this recommendation. Also, there is a stormwater pond designed to control runoff from the study area, does it help the N situation? (I know this is outside of the paper scope, but so are your comments on SW quality controls).

Reviewer #2: This article is well written and provides an interesting assessment of nitrogen species partitioning in urban runoff. The most compelling section was the isotope analysis based on its novelty. I have relatively minor comments on the attached document. I have a few items that should be cleared up for publication. In particular, I would ask the authors to specify if they checked for normality of the data.

Reviewer #3: General

The authors present an extensive and very valuable stormwater data set; in particular, the within-event isotopic analyses are a unique and interesting data set. The data have been thoroughly checked against previous studies and data sets. The field, lab, and statistical analyses are solid, and appropriate for the work. The paper is also pretty well written, logically organized, and thoroughly referenced. The results are a bit long and the conclusions a bit light, but mostly appropriate given the large size of the data set.

I have a few general comments that could potentially improve the clarity and impact of the results:

1) I think the conclusions could be strengthened by digging into the more novel aspects of the data set (DON, PON sources; within-event source dynamics) and making recommendations for BMPs or providing support for existing BMPs for treating those forms of N.

2) There is a lot of detail on the NO3 results (for good reason, given the extensive number of isotope samples that were processed), but it appears to be a minor component of TN at this site, and not a lot to be concluded from the results. NO3 and NH4 concentrations of 0.2mg/L (mean) are pretty low. I think the organic N part of the work is potentially more useful / interesting.

3) I would be interested in learning more about the within-event concentrations and isotopic composition, and what the results might mean for type or timing of stormwater management; and how might event characteristics (intensity, depth, duration), which vary across climates, impact prescribed management? I feel that this is a very important component of your data set!

4) “percent runoff” / “percent of runoff” needs to be more carefully defined for clarity (see comment below)

--

Minor Comments:

- check references to tables/figures so that they are consistent with the form required by the journal, e.g. “S4 Table” in L276 (should this be “Table S4?”), “Fig 3A and 4B” in L278 (should this be “Figs. 3A and 4B”), and elsewhere

- in citing studies, be consistent with locations (i.e. use City, State, Coutry or just State, Country)

Methods

- does the site have any baseflow? threshold for sampling seems small so guessing not but maybe good to mention

L135-144: check grammar (“enabled to monitor”… etc)

L146+: time paced sampling, or flow paced sampling? FWMC implies time pacing but is unclear.

- each sample bottle analyzed separately, right?

L160: is FWMC calculated for each event, or across all samples? If this is calculated across all samples, this would seem to bias results towards more frequently sampled storms (?)

L222: omit comma

Results

L233,L238: seems like a small threshold for runoff yet only 36% of rainfall and 30% of runoff was sampled — how applicable might results be for small storms?

L242: what quantities are being compared here — volumes? And if so, how is this different from the result presented in L248-9 (and Fig 3C)? Also, does the correlation include sampled events, or all events? Correlation of rainfall and runoff is an intuitive/obvious result, but this still needs to be explained.

L242-3 (and L332): what is “percent runoff” — fraction of rainfall that became runoff, or fraction of total runoff that was sampled, or something else? The first definition is implied by the subsequent lines, but an explicit definition would be helpful. This is also especially confusing because the y-axis label in Fig 3A (and x-axis in 3B) is “percent OF runoff”, which implies a potentially different definition (e.g. “percent of runoff that was observed”, rather than “percent of rainfall as runoff”). I think this term needs to be changed for clarity, even if is more verbose.

L247: similarly, 10-80% of what? rainfall?

L260: should be “Fig. S2”?

L282-4: check grammar

L333: course should be coarse

L342: are these expected to be major sources of water in your study area?

L363-4: this seems to conflict with the last sentence of the previous paragraph, which stated that rainfall and runoff samples were “identical with each other” — please clarify what difference is being referenced here (event-to-event perhaps?)

L370+, L352+, L501+: would it be useful to condense the isotope data (means and ranges) to a table, maybe along with references for similar studies? Your main points in these paragraphs (e.g. that runoff water was nearly all rainfall, explanation of seasonal and intra-event variability of dN-NO3) get buried at the end of the paragraphs. I think these would be easier to read and stronger paragraphs if you lead each with a main result from your study. Might be personal preference, though.

L393-395: this is a somewhat vague explanation of variability of isotopic composition within events — you provide a nice explanation in the previous paragraph for seasonal variability of dN-NO3, but perhaps you could do more with your data to explain how the changes in this isotope over an event explain changes in N source?

L397: I think this section could be expanded a bit to discuss changing sources during events, and if possible, discussing both NO3 and ON.

L403: the result beginning on L403 (change in source over the event) seems like a more interesting starting point than just summarizing your isotope results. Or could start a new paragraph here to discuss changes in source over an event. This is an important issue that your dataset is well equipped to address!

L410: this sentence needs clarification. Why would DIN become a larger component of TN as the larger events become “enriched” in ON? This would imply the opposite pattern. Do you perhaps mean “saturated” in ON, i.e. only so much can be mobilized?

413: Discussion of fertilizer kind of comes out of nowhere. Might be good here to point readers to Fig 5 again to show the importance of fertilizer — which appears to be important only for a few events?

424-499: I realize that the substantial amount of nitrate isotope data collected compels you to do a source-tracking analysis, which is interesting, but it comprises such a small component of stormwater N at your site, and the largest component is atmospheric, such that management implications are pretty minimal. The thorough discussion of the various components of atmospheric NO3 (L450-466) and organic NO3 (L481-99) do not add much to the main points of your paper and could potentially be condensed into a single paragraph discussing other NO3 sources.

- this section also needs to reference Fig 6

L467+: I think the fertilizer section might be the most important part of the NO3 results, especially in light of the fertilizer ban in place during the study period. How does the percentage of NO3/NH4 as fertilizer in your study compare to the other cited studies? If fertilizer restrictions were NOT in place in those other studies, then you have an interesting result for the effectiveness (or lack thereof) of a fertilizer ban.

L510, L517: this is an important result. It might be personal preference, but leading with a summary of the results is less exciting than leading with your major outcome. Also, oaks may be considered "messy" trees; how might the results change with a watershed containing different deciduous species (or mix of species)?

L520: Ong reference needs to be in numbered form

Conclusions

- Mostly just a summary of the results. What have you learned from the data set? (Most of the BMP suggestions in the conclusions are practices that are already being implemented for both N and P management.) For example, what insights did you gain from doing the within-event analyses, which are fairly rare as a data set?

L537: if DON is the dominant form, what might be some management options? The BMPs suggested for management of NO3 (L547-9), which is a minor component of TN, are already pretty standard practices, so this is not new information.

L554+: does this imply that street sweeping could be a particularly effective BMP?

--

Figures

Fig 2

- line graph implies continuity from point to point, yet these are discontinuous data (discrete x axis rather than continuous/time). Could this info be condensed into a table, or perhaps make all of the plots bar graphs (like top plot in figure)?

Fig 3

- A,B: see above concerns about “percent of runoff” label on y-axis

- C: is the line fit to all events or just sampled events?

Fig 4

- B: y-axis is really hard to read; possible to make text larger?

- legend: remove “for easy visualization” (which is subjective) and describe how they are grouped — it appears they are grouped by magnitude of TN (i.e. scale)?

Fig 6

- Legend: please give the dates of the events (rather than “May to September 2016”), and maybe rainfall amounts? Also please explain that “n” is number of samples within the event.

6. PLOS authors have the option to publish the peer review history of their article (what does this mean?). If published, this will include your full peer review and any attached files.

Reviewer #1: No

Reviewer #2: No

Reviewer #3: No

---

## [Author Response · Author response to Decision Letter 0]

17 Jan 2020

Please see attached response to reviewers comment file.

---

## [Editor Report · Decision Letter 1]

13 Feb 2020

Composition of Nitrogen in Urban Residential Stormwater Runoff: Concentrations, Loads, and Source Characterization of Nitrate and Organic Nitrogen

PONE-D-19-21012R1

Dear Dr. Toor,

We are pleased to inform you that your manuscript has been judged scientifically suitable for publication and will be formally accepted for publication once it complies with all outstanding technical requirements.

With kind regards,

Julian Aherne

Academic Editor

PLOS ONE

Additional Editor Comments (optional):

The authors have revised the manuscript in line with the comments from the three reviewers. I recommend that it is accepted for publication.
---

## [Editor Report · Acceptance letter]

19 Feb 2020

PONE-D-19-21012R1 

Composition of Nitrogen in Urban Residential Stormwater Runoff: Concentrations, Loads, and Source Characterization of Nitrate and Organic Nitrogen 

Dear Dr. Toor:

I am pleased to inform you that your manuscript has been deemed suitable for publication in PLOS ONE. Congratulations! Your manuscript is now with our production department. 

With kind regards,

on behalf of

Dr. Julian Aherne 

Academic Editor

PLOS ONE